# Luxury or Cultural Tourism Activities? The Role of Narcissistic Personality Traits and Travel-Related Motivations

**DOI:** 10.3390/bs14100972

**Published:** 2024-10-20

**Authors:** Avi Besser, Villy Abraham, Virgil Zeigler-Hill

**Affiliations:** 1Department of Communication Disorders, Hadassah Academic College, Jerusalem 9101001, Israel; 2Department of Technological Marketing, Sapir Academic College, “Shaar HaNegev” Educational Campus, Ashkelon Beach 7915600, Israel; abraham.villy@gmail.com; 3Department of Psychology, Oakland University, Rochester, MI 48306, USA

**Keywords:** narcissistic personality traits, travel motivations, luxury tourism, cultural tourism, activity preferences, tourism management

## Abstract

This study aimed to examine the role of travel-related motivations and preferences for activities (such as learning about local culture, relaxation and entertainment, status and social recognition, escape and curiosity, and experience) in the relationship between narcissistic personality traits and the preference for luxury tourism versus cultural tourism. A sample of 1342 Israeli community members was analyzed. The methodology utilized parallel mediation analyses to assess how different forms of narcissism (extraverted, antagonistic, neurotic, and communal) were associated with the desire for luxury tourism over cultural tourism, as mediated by various travel-related motivations. The results indicated that narcissistic personality traits were generally associated with a preference for luxury tourism, primarily through indirect associations via travel motivations such as relaxation and entertainment, status and social recognition, and experience. The conclusion highlights the implications for understanding how specific travel-related motivations influence tourism preferences among individuals with varying levels of narcissistic traits, suggesting that tourism management strategies could benefit from a nuanced approach to personality in marketing.

## 1. Introduction

Travel holds significant value in people’s lives due to the satisfaction and pleasure they derive from such activities. Despite its importance, psychology has not extensively considered issues surrounding travel. However, recent research has started exploring how personality traits influence individuals’ approaches to travel. Traits such as extraversion, neuroticism, and sensation-seeking have been linked to various aspects of travel, including expectations, stress levels, and overall satisfaction [1,2,3,4,5,6]. Moreover, narcissistic personality traits have emerged as significant factors in both anticipating and experiencing travel-related events [7].

This study aimed to explore whether narcissistic traits influenced preferences for specific types of travel (e.g., luxurious travel). Furthermore, this study sought to investigate whether motives related to travel (e.g., the motivation to gain status and social recognition) mediated the relationships that narcissistic traits had with travel-related preferences.

The contemporary luxury market is witnessing a significant shift toward experiential luxury, exemplified by premium food and beverages, bespoke vacations, and luxurious accommodations [8,9,10]. This transformation underscores a fundamental change in luxury consumption, where the emphasis is placed on experiences rather than material possessions. Consequently, scholarly interest has grown in exploring both luxury experiences and luxury tourism [11,12]. Although the recent literature reviews have extensively examined luxury services in a broader context [13], there remains a need for focused research into the dynamics of experiential luxury and its implications for the luxury tourism sector [11,12].

### Objectives

Despite the attention given to luxury travel, existing theories regarding consumer behaviors in this domain remain relatively underdeveloped [14]. Some studies suggest that luxury travel serves as a means for self-enhancement, allowing travelers to cultivate an idealized self-image and make impressions on others through their luxurious experiences [15,16]. However, less emphasis has been placed on exploring the potential connections that personality traits may have with travelers’ inclinations toward luxury tourism and the underlying motivations. This study aims to address this gap by examining the associations that narcissistic personality traits have with the preference for luxury tourism over cultural tourism, considering travel-related motivations as potential mediating factors.

## 2. Literature Review

Luxury tourism is typically defined by its emphasis on high-end accommodations, personalized services, and exclusive experiences [17]. Travelers engaged in luxury tourism seek indulgence, relaxation, and sophistication, with a focus on premium amenities and impeccable service standards [11,12,18]. In contrast, cultural tourism revolves around the exploration of heritage sites, museums, festivals, and local traditions [19]. It is characterized by a desire for authentic encounters, immersive learning experiences, and meaningful interactions with host communities [18]. Travelers engaged in cultural tourism often prioritize educational enrichment, cultural exchange, and sustainability [20]. Despite these differences, both forms of tourism cater to travelers who prioritize unique and immersive experiences, offering distinct yet enriching opportunities.

Since the 1990s, cultural tourism has emerged as a widespread phenomenon, encompassing various distinct categories including heritage, arts, gastronomy, film, and creativity [21]. This growth trend has been consistent, with the luxury tourism and hospitality sector experiencing remarkable expansion, valued at USD 1.3 billion in 2022, with projections indicating further growth to reach a value of USD 2.7 billion by 2032, expanding at a compound annual growth rate of 7.8% between 2023 and 2032 [22,23]. Recent research has delved into various aspects of luxury tourism, including the perceived value and characteristics of luxury travel experiences, factors contributing to satisfaction with luxury hotels, the connection between brand equity and the financial performance of luxury travel providers, and the efficacy of advertising strategies in enhancing consumer attitudes toward luxury travel providers [14,16,24,25,26,27,28]. Although luxury and cultural tourism represent distinct segments of the tourism industry, they are complementary. That is, both types of tourism focus on providing unique and enriching experiences to travelers that go beyond conventional tourism and both cater to travelers seeking exclusive and immersive experiences that leave a lasting impact [29]. Luxury tourism emphasizes opulence and exclusivity, whereas cultural tourism prioritizes authenticity and cultural immersion. Understanding these differences is crucial for destination stakeholders to develop targeted marketing strategies, enhance visitor experiences, and maximize socio-economic benefits.

One crucial aspect in understanding traveler preferences lies in their travel motivations, which have been shown to profoundly impact destination selection and subsequent behavior [30]. Motives are distinct internal inclinations that stimulate, guide, and unify an individual’s actions in alignment with their needs, values, or attitudes. They are commonly acknowledged as precursors to behavior [31,32]. Within leisure studies, the interconnections among these elements are often conceptualized as a sequential process, wherein needs, preferences, motives, desires, and expectations collectively influence consumer behavior or engagement in activities [33,34]. This link between travel motivation and destination choice revolves around the perceived benefits travelers associate with a specific destination [35]. Song and Bae (2018) assert that various motives contribute to the travel decision-making process, albeit with discrepancies in their relative significance [36]. For instance, Ward (2014) demonstrated the multifaceted interests and motivations and their connections with destination preferences [37].

It has been shown that outdoor activities, immersion in nature, picturesque landscapes, and opportunities for relaxation emerge as crucial motivators for selecting rural tourism destinations [38]. In cultural tourism, Chang et al. (2020) highlighted the desire to explore new cultures and the aspiration to deviate from mundane dietary routines and enhance social status among peers as significant motivations for culinary tourists in choosing specific destinations [39]. Moscardo et al. (2016) emphasized activities’ crucial role in the bonds between travelers and their destinations. They suggest that activities are central aspects of destinations, judged by travelers according to their perceived capacity to satisfy personal needs [40]. Consequently, motives can be seen as shaping travelers’ expectations regarding activities, with destinations viewed as providing avenues for participating in those activities. As a result, travel motivations are intricately intertwined with preferences for activities, thereby impacting destination choices [40].

Recent research has identified five distinct tourism motivation factors [41]: learning about the local culture, relaxation and entertainment, social status and recognition, escape and curiosity, and experience. Cultural learning emerges as a pivotal motivation for engaging in cultural tourism [42,43]. Similarly, relaxation and entertainment are consistently reported in other studies [44]. The desire for travel bragging and self-esteem aligns with the factor of social status and recognition and has been recognized as a significant motivator for cultural tourism [45]. Additionally, escapism and curiosity have been identified as important factors in prior research [46,47]. Lastly, the pursuit of unique experiences resonates with findings in the existing literature [48].

The Theory of Planned Behavior (TPB; [49]) asserts that intentions are shaped by individuals’ inclination to engage in specific behaviors, influenced by attitudes, subjective norms, and perceived behavioral control [49]. Hsu and Huang (2010) extended the original TPB model to the study of tourism by including the motivating factors affecting individuals’ travel decisions, attitudes, and the impact of various reference groups on tourists’ intentions to visit a destination [50]. In their proposed framework, the motivation and attitude toward visiting a destination are directly associated with individuals’ behavioral intentions which are, in turn, associated with actual behaviors regarding visiting the destination [51].

Moreover, individual personality traits, particularly narcissism, may play a role in shaping preferences within the luxury tourism market. In recent years, there has been a growing consensus that narcissism is a nuanced construct with multiple facets rather than a singular and uniform concept [52]. One model of narcissism posits three core traits: extraverted narcissism, characterized by exhibitionism and self-assurance; antagonistic narcissism, marked by defensiveness and aggression; and neurotic narcissism, typified by distress and a strong desire for approval [53,54]. This recognition of the multidimensionality of narcissism has led researchers to explore the conflicting trends observed in the literature. For instance, while the grandiose aspects of narcissism (e.g., extraverted narcissism) often show a positive attitude towards wealth and luxury items, viewing them as tools for asserting dominance, the vulnerable aspects of narcissism (e.g., neurotic narcissism) tend to have more ambivalent associations, seeing them as desirable yet a source of anxiety [55,56]. These findings highlight the potential importance of distinguishing between different facets of narcissism when studying its relationship with travel-related preferences.

Although communal narcissism—seeking admiration for perceived altruism and selflessness—is not explicitly addressed in the traditional model of narcissism, it has gained significant attention in recent years [57,58]. Including communal narcissism alongside extraverted, antagonistic, and neurotic narcissism is crucial for a comprehensive understanding of narcissistic personality traits and their implications. Although extraverted, antagonistic, and neurotic narcissism have received considerable attention due to their disruptive manifestations, communal narcissism represents a distinct and relatively understudied facet of narcissism. Studying communal narcissism alongside traditional dimensions broadens the understanding of narcissism’s multifaceted nature and diverse behavioral outcomes. Moreover, communal narcissism’s focus on seemingly prosocial traits such as generosity and kindness may offer valuable insights into how narcissistic individuals view tourism issues as opportunities to showcase their perceived communal virtues.

Therefore, considering communal narcissism alongside other narcissistic traits has the potential to enrich the understanding of the connections that narcissism has with preferences for tourism. For instance, individuals high in communal narcissism may particularly enjoy participating in cultural tourism, as it allows them to receive admiration for their nonmaterialistic preferences. Moreover, research often delves into the social and personal advantages associated with luxury travel, including self-enhancement, prestige, brand loyalty, social status, and superior quality [14,15,16,24,28,59]. Furthermore, there is a body of research considering luxury tourism as part of luxury consumption or conspicuous consumption [16,28], and according to Kim (2018), “Consumers can engage in luxury experiential consumption to satisfy the desire for exclusivity, in addition to the desire to live well and engage in a wide range of valuable moments” (p. 280) [60]. Since narcissism is associated with conspicuous consumption [61,62], it seems reasonable to consider the possibility that narcissistic personality traits may also be associated with preferences for luxury tourism vs. cultural tourism.

Narcissism is a multifaceted trait characterized by self-preoccupation, grandiosity, entitlement, a constant need for admiration, and a lack of empathy [63], which often leads individuals to seek validation and approval through various means, including displaying luxury possessions [64,65]. Narcissistic tendencies are linked to a preference for wealth, social status, and material possessions [66,67,68]. The pursuit of luxury items may serve as a tool for narcissists to garner admiration and external validation [62,69,70], with the need for approval acting as a driving force behind conspicuous consumption behaviors [55,62,71]. The desire for uniqueness also plays a significant role, as narcissistic individuals strive for positive distinctiveness, leading them to opt for unconventional and personalized luxury products [61,62,71]. In addition, it has been found that sharing on social media networking and social media envy drive millennials’ aspirational tourism consumption [72], which might also motivate narcissists to prefer luxury tourism activities. Additionally, materialism emerges as a mediator in the relationship between narcissism and conspicuous consumption, as narcissists value material possessions as symbols of status and success [56,61].

Based on the reviewed literature and the identified gaps, the following were hypothesized:

**Hypothesis 1:** Narcissistic personality traits will be positively associated with preferences for luxury tourism over cultural tourism. This hypothesis is based on studies suggesting that individuals with high levels of narcissism are motivated by a desire for status and social recognition, which luxury tourism provides more than cultural tourism [14,15].

**Hypothesis 2:** Travel-related motivations (e.g., the motivation for social status and recognition) will mediate the associations that narcissistic traits have with preferences for luxury tourism. This hypothesis is based on prior research showing that motives significantly influence travel decisions among individuals with narcissistic tendencies, thereby affecting their tourism preferences [24,29,59].

**Hypothesis 3:** Extraverted narcissism will be positively correlated with preferences for luxury tourism. This prediction stems from evidence that individuals exhibiting extraverted traits often engage in conspicuous consumption and prefer experiences that are prominent and attention-grabbing, which is typical of luxury tourism [52,53].

**Hypothesis 4:** Antagonistic narcissism will be negatively associated with cultural tourism preferences. Individuals high in antagonistic traits may favor luxury experiences that satisfy their self-serving needs rather than engaging in the community-oriented and cooperative nature of cultural tourism [55,56].

**Hypothesis 5:** Neurotic narcissism will have complex associations with both luxury and cultural tourism preferences, characterized by ambivalence due to underlying anxieties about self-image and social acceptance. This may lead to inconsistent tourism choices based on the context and perceived judgment from others [55,56].

**Hypothesis 6:** Communal narcissism will be positively associated with preferences for cultural tourism, as individuals high in communal narcissism may be driven by a desire to showcase their perceived altruistic traits through cultural engagement, thereby seeking admiration from others [57,58].

**Hypothesis 7:** The desire for social status and recognition will positively mediate the associations that narcissistic personality traits have with luxury tourism preferences. This reflects the tendency for narcissistic individuals to prioritize prestige in their travel choices, which aligns with previous findings that emphasize the role of social status motives in tourism behavior [24,28,59] and the relationship between narcissism and luxury tourism preferences. Narcissistic individuals may value luxury items as symbols of status and success, which in turn influences their tourism preferences [62,66,67,68].

Figure 1 presents the proposed associations that narcissistic personality traits may have with the preference for luxury tourism over cultural tourism activities through tourist motives.

## 3. Materials and Methods

### 3.1. Participants and Procedure

This study included a convenience sample of 1527 members of the Israeli community who volunteered to participate by responding to requests distributed through flyers in public areas and postings on various social media platforms. All questionnaires used in the present study were administered in Hebrew after being translated from the original English versions using the back-translation method. Data for 185 participants were excluded due to reasons such as failing two or more attention checks or having invariant response patterns (i.e., selecting the same response for a large percentage of the items). Out of the initial participants, the final sample consisted of 1342 individuals (728 men and 614 women) with an average age of 40.50 years (*SD* = 14.80, ranging from 19 to 86 years). On average, participants reported 15.08 years of formal education (*SD* = 3.03), with the following current employment situation: 62% working full-time, 18% working part-time, 8% going to school, 6% retired, 3% unemployed, and 3% other. Most participants identified as Jewish (97%), with 57% being married, 22% being single, 13% exclusively dating, 7% divorced, and 1% widowed. Regarding their self-reported current economic status, 44% described themselves as “below average”, 24% described themselves as “average”, and 32% described themselves as “above average”. Finally, on average, participants reported that they took vacations 2.03 times a year (SD = 1.54, ranging from 0 to 20). This study was not pre-registered, but the data file is available on the Open Science Framework (OSF) at: https://osf.io/d4akt/.

### 3.2. Questionnaires

#### 3.2.1. Narcissism

The short form of the *Five-Factor Narcissism Inventory* [73] was employed to assess extraverted narcissism (16 items; “I like being noticed by others” [α = 0.80]), antagonistic narcissism (32 items; “I hate being criticized so much that I can’t control my temper when it happens” [α = 0.87]), and neurotic narcissism (12 items; “When people criticize me, I get embarrassed” [α = 0.82]). Responses were provided using scales that ranged from “*strongly disagree*” (1) to “*strongly agree*” (5).

#### 3.2.2. Communal Narcissism

The *Communal Narcissism Inventory* [57] was used to capture communal narcissism (16 items, e.g., “I am the most helpful person I know” [α = 0.90]). Responses were provided using scales that ranged from “*strongly disagree*” (1) to “*strongly agree*” (5).

#### 3.2.3. Tourist Motives

*The Tourist Motives Scale* [33] was used to measure the following motives for participating in tourism: learning about the local culture (5 items; “To learn about local history” [α = 0.73]), relaxation and entertainment (7 items; “To have thrills and excitement” [α = 0.63]), status and social recognition (4 items; “To gain the respect of others” [α = 0.71]), escape and curiosity (4 items; “To satisfy my curiosity” [α = 0.60]), and experience (7 items; “To see famous cultural places” [α = 0.70]) [41]. Responses were provided using scales that ranged from “*strongly disagree*” (1) to “*strongly agree*” (5).

#### 3.2.4. Preference for Luxury Tourism over Cultural Tourism

Participants were provided with descriptions of two types of vacations. Both vacations included a series of tourist destinations and activities.

*(A). “This vacation includes: guided cultural-focused walking tours, cultural tours, visiting arts and crafts stores, and attending cultural festivals or events. It also includes township tourism, visits to traditional villages, homestays, and medicinal plant tours. The vacation also includes visiting museums, architectural and archaeological sites, as well as historical or heritage sites, and landmarks”*.

*(B) “This vacation includes: a visit to a fancy resort, visiting an expensive spa and receiving treatments, dining in exclusive restaurants, and visiting casinos. Also, the vacation includes going skiing and playing golf, visiting wineries, and taking a pleasure cruise on a yacht. The vacation includes visiting exotic destinations with beautiful views, staying in luxury hotels, visiting malls and shopping centers where you can shop for exclusive brands, and experiencing special attractions (e.g., driving in a luxury car, scuba diving, bungee jumping)”*.

Participants were asked to rate these two vignettes using a series of items. First, they were asked to indicate their *general interest* in each type of vacation: “How interested would you be in going on a vacation that includes the activities described”? on a seven-point Likert scale ranging from “*very much*” (1) to “*not at all* (7). Then, they were asked to indicate their intentions and attitudes, adapted from Carfora et al., 2019 [74]. For *intentions*, the following 3 items were rated with a seven-point Likert scale ranging from “*strongly disagree*” (1) to “*strongly agree*” (7): “*I intend to visit a holiday destination that includes the described activities*”, “*I plan to visit a holiday destination that includes the described activities*”, and “I *want to visit a holiday destination that includes the activities described above*”. For *attitudes*, the following 5 items were rated using a five-point Likert scale: “*Visiting a holiday destination that includes the activities described above is*” [*bad* (1) to *good* (5), *harmful* (1) to *beneficial* (5), *unpleasant* (1) to *pleasant* (5), *unenjoyable* (1) to *enjoyable* (5), and (1) *foolish* to (5) *wise*]. Standardized composites were created for their ratings of each vignette and then a difference score between these ratings was computed, such that higher scores reflected a stronger preference for luxury tourism over cultural tourism.

### 3.3. Ethics Statement

Participation in this study was voluntary, and participants were aware that they could withdraw from the study at any time. All participants provided their signed, informed consent. No social security numbers or other identifying data were collected, nor were any invasive examinations conducted. This project was conducted with the approval of the Ethics Committee (IRB) of Hadassah Academic College.

### 3.4. Statistical Analysis

First, a (2) × (3) repeated measures MANOVA was performed to assess differences in positivity measures (general interest, intentions, and attitudes) between cultural and luxury tourism. Next, correlation coefficients were calculated to examine the zero-order relationships between narcissistic personality traits, tourist motivations, and the preference for luxury tourism over cultural tourism. These analyses were conducted using SPSS version 26 (SPSS Inc., Chicago, IL, USA). Following the correlation analysis, a series of parallel multiple mediation analyses were performed using the PROCESS macro [75], controlling for demographic factors such as age, gender, education, and income. These analyses were conducted to test the hypotheses that narcissistic personality traits would be linked to tourist motivations, which would, in turn, influence the preference for luxury tourism over cultural tourism. This approach allowed us to explore whether narcissistic traits had indirect associations with the preference for luxury tourism through tourist motivations. Specifically, narcissism was used as the predictor, tourist motivations (i.e., learning about local culture, relaxation and entertainment, status and social recognition, escape and curiosity, and experiences) were the mediators, and the preference for luxury tourism over cultural tourism was the outcome variable. Due to the overlap among narcissistic traits, separate analyses were conducted for each trait, with each serving as the predictor in its own model. For all statistical tests, two-tailed significance tests were applied, and confidence intervals were based on *p* < 0.05.

## 4. Results

### 4.1. Comparisons of the Levels of Positivity Measures (General Interest, Intentions, and Attitudes) for Cultural and Luxury Tourism

A (2) × (3) repeated measures MANOVA with two levels within the subject type of the vignette (Cultural and Luxury) and three measures of the rate of the level of Positivity for each of these two vignettes (Positive General Interest, Positive Intentions, and Positive Attitudes) was conducted. The M and SD for general interest, positive intentions, and positive attitudes toward luxury tourism over cultural tourism are presented in Table 1.

The results indicated a significant effect for the Type of Vignette (*F* _[1,1341]_ = 15.29, *p* < 0.001), with significantly overall higher positive rates for luxury compared to cultural tourism. In addition, the significant effect obtained for the positivity measures (*F* _[2,1341]_ = 520.11, *p* < 0.001) indicates significant differences in their levels, which were qualified by the significant Type of Vignette × Positivity Measures (*F* _[2,1341]_ = 86.67, *p* < 0.001). As can be seen in Table 1, for luxury tourism, the interest and intentions are significantly higher than for cultural tourism, while the attitudes are higher for cultural tourism. This interaction is presented in Figure 2.

### 4.2. Univariate Analyses

The correlation coefficients and descriptive statistics are presented in Table 2. Extraverted narcissism had positive correlations with the motives for relaxation and entertainment, status and social recognition, escape and curiosity, and experience, which ranged from small to large in magnitude, but it was not correlated with a motivation to learn about the local culture. Antagonistic narcissism positively correlated with motives for learning about the local culture, relaxation and entertainment, status and social recognition, escape and curiosity, and experience, which ranged from very small to very large in magnitude. Neurotic narcissism had very small positive correlations with motives for relaxation and entertainment, status and social recognition, escape and curiosity, and experience, which ranged from small to large in magnitude, whereas it had a very small *negative* correlation with the motivation for learning about the local culture. Communal narcissism positively correlated with the motives for learning about the local culture, relaxation and entertainment, status and social recognition, escape and curiosity, and experience, which were small to large in magnitude.

Extraverted narcissism, antagonistic narcissism, and communal narcissism had small to medium positive correlations with the preference for luxury tourism over cultural tourism, whereas neurotic narcissism was not correlated with this preference. This pattern partially supports Hypothesis 1. The motives for relaxation and entertainment, status and social recognition, escape and curiosity, and experience had very small to medium positive correlations with the preference for luxury tourism over cultural tourism. This supports the essential requirements [76] for testing Hypothesis 2. In contrast, the motivation for learning about the local culture had a large *negative* correlation with the preference for luxury tourism over cultural tourism.

### 4.3. Multivariate Analyses

The results of the parallel multiple mediation analyses are presented in Table 3. Extraverted narcissism had a positive total association with the preference for luxury tourism over cultural tourism (*β* = 0.17, *SE* = 0.03, *t* = 6.77, *p* < 0.001, *CI*_95%_ [0.12, 0.22]) that persisted when the mediators were included in the model (*β* = 0.11, *SE* = 0.02, *t* = 4.89, *p* < 0.001, *CI*_95%_ [0.07, 0.16]). This supports Hypothesis 3. Extraverted narcissism had positive indirect associations with the preference for luxury tourism over cultural tourism through the motives for relaxation and entertainment (*β* = 0.06, *SE* = 0.01, *z* = 5.47, *p* < 0.001, *CI*_95%_ [0.04, 0.08]), status and social recognition (*β* = 0.03, *SE* = 0.01, *z* = 3.38, *p* < 0.001, *CI*_95%_ [0.01,0.04]), and experience (*β* = 0.03, *SE* = 0.01, *z* = 4.40, *p* < 0.001, *CI*_95%_ [0.02, 0.05]). In contrast, extraverted narcissism had a negative indirect association with the preference for luxury tourism over cultural tourism through the motive for learning about the local culture (*β* = −0.06, *SE* = 0.01, *z* = −4.32, *p* < 0.001, *CI*_95%_ [−0.09, −0.03]), whereas it did not have an indirect association with the preference for luxury tourism over cultural tourism through the motive for escape and curiosity (*β* = 0.00, *SE* = 0.00, *z* = −0.46, *p* = 0.645, *CI*_95%_ [−0.01, 0.01]). This partially supports Hypothesis 2 and provides support for Hypothesis 7.

Antagonistic narcissism had a positive total association with the preference for luxury tourism over cultural tourism (*β* = 0.13, *SE* = 0.03, *t* = 5.28, *p* < 0.001, *CI*_95%_ [0.08, 0.18]) that persisted when the mediators were included in the model (*β* = 0.10, *SE* = 0.02, *t* = 4.06, *p* < 0.001, *CI*_95%_ [0.05, 0.15]). This does not support Hypothesis 4. Antagonistic narcissism had positive indirect associations with the preference for luxury tourism over cultural tourism through the motives for relaxation and entertainment (*β* = 0.02, *SE* = 0.01, *z* = 3.42, *p* < 0.001, *CI*_95%_ [0.01, 0.04]), status and social recognition (*β* = 0.03, *SE* = 0.01, *z* = 2.88, *p* = 0.004, *CI*_95%_ [0.01, 0.06]), and experience (*β* = 0.04, *SE* = 0.01, *z* = 4.29, *p* < 0.001, *CI*_95%_ [0.02, 0.06]). In contrast, antagonistic narcissism had a *negative* indirect association with the preference for luxury tourism over cultural tourism through the motive for learning about the local culture (*β* = −0.06, *SE* = 0.01, *z* = −4.45, *p* < 0.001, *CI*_95%_ [0.08, −0.03]). Antagonistic narcissism was not indirectly associated with the preference for luxury tourism over cultural tourism through the motive for escape and curiosity (*β* = 0.00, *SE* = 0.00, *z* = −1.03, *p* = 0.304, *CI*_95%_ [−0.02, 0.00]). This partially supports Hypothesis 2, offers more complex insight into Hypothesis 4, and provides support for Hypothesis 7.

Neurotic narcissism did not have a total association with the preference for luxury tourism over cultural tourism (*β* = 0.02, *SE* = 0.02, *t* = 0.65, *p* = 0.517, *CI*_95%_ [−0.03, 0.06]), but a *negative* association emerged when the mediators were included in the model (*β* = −0.05, *SE* = 0.02, *t* = −2.45, *p* = 0.014, *CI*_95%_ [−0.10, −0.01]). This supports Hypothesis 5. Neurotic narcissism had positive indirect associations with the preference for luxury tourism over cultural tourism through the motives for relaxation and entertainment (*β* = 0.02, *SE* = 0.01, *z* = 2.94, *p* = 0.003, *CI*_95%_ [0.01, 0.03]), status and social recognition (*β* = 0.02, *SE* = 0.01, *z* = 3.87, *p* < 0.001, *CI*_95%_ [0.01, 0.03]), and experience (*β* = 0.02, *SE* = 0.01, *z* = 3.03, *p* = 0.002, *CI*_95%_ [0.01, 0.03]). In contrast, neurotic narcissism did not have an indirect associations with the preference for luxury tourism over cultural tourism through the motives for learning about the local culture (*β* = 0.01, *SE* = 0.01, *z* = 1.12, *p* = 0.264, *CI*_95%_ [−0.01, 0.04]) or escape and curiosity (*β* = 0.00, *SE* = 0.00, *z* = −0.32, *p* = 0.749, *CI*_95%_ [−0.01, 0.01]). This offers more complex insight into Hypotheses 2 and 5 as well as support for Hypothesis 7.

Communal narcissism had a positive total association with the preference for luxury tourism over cultural tourism (*β* = 0.10, *SE* = 0.02, *t* = 3.83, *p* < 0.001, *CI*_95%_ [0.05, 0.14]) that persisted when the mediators were included in the model (*β* = 0.09, *SE* = 0.02, *t* = 3.83, *p* < 0.001, *CI*_95%_ [0.04, 0.13]). This supports Hypothesis 6. Communal narcissism had positive indirect associations with the preference for luxury tourism over cultural tourism through the motives for relaxation and entertainment (*β* = 0.05, *SE* = 0.01, *z* = 5.55, *p* < 0.001, *CI*_95%_ [0.03, 0.07]), status and social recognition (*β* = 0.03, *SE* = 0.01, *z* = 3.46, *p* < 0.001, *CI*_95%_ [0.01, 0.05]), and experience (*β* = 0.04, *SE* = 0.01, *z* = 4.58, *p* < 0.001, *CI*_95%_ [0.02, 0.06]). In contrast, communal narcissism had a *negative* indirect association with the preference for luxury tourism over cultural tourism through the motive for learning about the local culture (*β* = −0.11, *SE* = 0.01, *z* = −8.01, *p* < 0.001, *CI*_95%_ [−0.14, −0.08]). Communal narcissism was not indirectly associated with the preference for luxury tourism over cultural tourism through the motive for escape and curiosity (*β* = 0.00, *SE* = 0.01, *z* = −0.61, *p* = 0.544, *CI*_95%_ [−0.01, 0.01]). This partially supports Hypothesis 2, offers more complex insight into Hypothesis 6, and provides support for Hypothesis 7.

## 5. Discussion

The existing literature on luxury travel consumer behavior often overlooks the impact of individual personality differences on travel preferences and motivations. Building on previous studies that suggest luxury travel serves as a means for self-enhancement—allowing travelers to create an idealized self-image and impress others through their lavish experiences [15,16]—the present study aims to address this gap. The aim of the present study was to examine the associations that narcissistic personality traits had with the preference for luxury tourism over cultural tourism, focusing on the role of travel-related motivations as potential mediators.

The present study compared perceptions of luxury and cultural tourism in terms of general interest, intentions, and attitudes. The findings indicate that luxury tourism was perceived more favorably than cultural tourism in terms of general interest and intentions, but cultural tourism was perceived more favorably than luxury tourism in terms of attitudes. This complex pattern aligns with the existing literature on luxury tourism, which often portrays it as a product synonymous with high quality, exclusivity, and tailored top-tier services and dining experiences. Within the realm of luxury tourism, there is an emphasis on providing unique benefits and creating memorable moments for consumers [77]. The current study supports the notion that there is a shift among consumers from mere product consumption to a more experiential form of luxury consumption [78]. When juxtaposed with cultural tourism, luxury tourism is commonly perceived as elitist, exclusive, symbolic, prestigious, and expensive, with a strong emphasis on the products and their associated attributes. In this light, consumers derive value in the form of status through public symbolism [12].

However, despite the general interest in luxury tourism and the intentions to engage in luxury tourism, results revealed that attitudes towards luxury vacations were significantly lower in comparison to cultural vacations. Despite the allure of luxury getaways, individuals acknowledged the value of cultural vacations, deeming them as more enriching, more enjoyable, and wiser choices.

The results regarding the link between narcissistic personality traits and the preference for luxury tourism over cultural tourism were largely as anticipated. Extraverted narcissism, antagonistic narcissism, and communal narcissism exhibited similar positive connections with favoring luxury tourism over cultural tourism. These findings align with the existing literature indicating that luxury travel can serve as a way for individuals to self-enhance, enabling travelers to craft an idealized self-image and leave an impression on others through their extravagant experiences [13,14]. Furthermore, these results correspond with research highlighting the social and personal benefits linked to luxury travel, such as self-enhancement, prestige, brand loyalty, social status, and superior quality [14,15,16,24,28,59]. They also build upon prior findings that have linked narcissism with conspicuous consumption [61,62]. This association extends to include preferences for luxury tourism over cultural tourism.

Individuals with heightened narcissistic traits often strive for validation and approval through different avenues, such as showcasing luxury possessions [64,65]. They tend to gravitate toward symbols of wealth, social status, and material belongings [66,67,68]. Their desire for luxury items can be seen as a method to attract admiration and external validation [62,69,70]. Moreover, individuals with these traits possess a strong yearning for uniqueness and aim for positive distinctiveness, which compels them to select unconventional and personalized luxury products [61,62,71]. The findings of the present study extend these findings to include luxury tourism experiences as another way for individuals with narcissistic personality traits to positively distinguish themselves from others.

In contrast to the other narcissistic personality traits, neurotic narcissism had a negative association with the preference for luxury tourism over cultural tourism, but this association only emerged when controlling for the mediators. This observation could hint at a “suppression effect” within the mediation model [79]. This shift could signify the possibility that individuals with high levels of neurotic narcissism, who are often marked by distress and a pronounced yearning for approval [53,54], may view vacations, specifically luxury vacations, as stress-inducing experiences.

The narcissistic personality traits often had positive correlations with various travel motives. However, there were exceptions to this pattern. For example, neurotic narcissism exhibited a small negative correlation with the motivation to learn about local culture, whereas extraverted narcissism was not associated with this particular motive. One potential explanation for this pattern could be that narcissistic personality traits are similar in terms of many travel-related motives (e.g., the motivation to use travel as a way to gain status and recognition from others), but they diverge with regard to the extent to which they are motivated by the opportunity to engage in cultural exploration. Hence, these results suggest intriguing similarities and differences in the travel-related motives that characterize these narcissistic personality traits.

In the present study, all four types of narcissism had positive indirect associations with the preference for luxury tourism over cultural tourism through the motives for relaxation and entertainment, status and social recognition, and experience. This finding aligns with Hsu and Huang’s (2010) expansion of the TBP [33] within the realm of tourism research. The present study underscores the significance of considering the motivating factors that influence individuals’ travel choices and attitudes [49] and intentions to visit a destination [50], but also emphasizes the importance of examining predispositions of individual differences in personality traits when studying these dynamics.

The only indirect associations that differed for the narcissistic personality traits were through learning about the local culture where these indirect associations were negative for extraverted narcissism, antagonistic narcissism, and communal narcissism, whereas there was no indirect association for neurotic narcissism. The negative indirect associations that emerged for extraverted, antagonistic, and communal narcissism through learning about the local culture diverged from the positive indirect associations these forms of narcissism had with the preference for luxury travel through other travel-related motives. These conflicting indirect associations suggest that issues surrounding cultural engagement may be appealing to individuals with these personality traits, which may detract from their overall preference for luxury travel. This pattern aligns with the expectation that individuals with high levels of extraverted narcissism and communal narcissism would be motivated to learn about other cultures because this may serve as a form of virtue signaling that allows them to gain status by indicating their sophistication or expressing their prosocial characteristics. In contrast, this sort of pattern was not expected to emerge for antagonistic narcissism because it is marked by defensiveness and aggression [53,54], which has been assumed to limit the extent to which individuals would be motivated to invest in learning about local cultures. It will be important for future research in this area to clarify the exact role that the motivation to learn more about local cultures plays in the connection between antagonistic narcissism and travel-related preferences. For example, it is possible that individuals with high levels of antagonistic narcissism may be interested in learning more about other cultures to establish the superiority of their own culture over others.

These findings might have some implications for the field of tourism management. For example, the mediating role of the motivation to learn about the local culture in the relationship between narcissism and the preference for luxury tourism over cultural tourism points to this motivation’s potential leverage as a ‘pull’ factor that destinations can use to appeal to narcissistic tourists. Hence, destination management offices, tourism ministries, tourism managers, and others will likely benefit from emphasizing cultural aspects associated with the promoted destination. Nevertheless, the study findings point to the potential marketing benefits of distinguishing between various forms of narcissism, particularly concerning the mediating role of various motivating factors. The findings of this study suggest that learning about the local culture may not always positively influence a preference for luxury over cultural tourism. It may harm the latter, such as in the case of antagonistic and communal narcissism. Thus, distinguishing between different forms of narcissism to create distinct marketing campaigns emphasizing ‘pull’ motivation factors known to be most appealing to the target market cannot be overstated.

The finding that attitudes toward luxury tourism are less positive than attitudes toward cultural tourism suggests that destination management offices, tourism ministries, and tourism managers may benefit from focusing marketing campaigns on the early stages of the Consumer’s Buying Decision Process, particularly the “search for information stage” [80]. This is the stage where attitudes are formed as tourists are informed about a destination through social media, the internet, family, friends, or other sources [81]. The findings of the present study suggest that various forms of narcissism are likely to be associated with different motives. Hence, marketing campaigns are more likely to be successful if several are created, each targeting a segment distinguished by a particular form of narcissism.

A limitation of this research is the reliance on cross-sectional, self-reported data to examine mediational hypotheses. Although the results suggest that narcissistic traits are indirectly associated with a preference for luxury tourism over cultural tourism through specific motives and activity preferences, the cross-sectional nature of the data prevents us from drawing firm conclusions about causality. For example, it is possible that a sustained desire for status and social recognition contributes to the development of narcissistic traits, rather than narcissism driving these motives. To better understand these causal dynamics, future studies should utilize experimental or longitudinal methods.

Additionally, the study’s sample was drawn from a single cultural group (Israel), which limits the generalizability of the findings. Cultural differences could influence the relationships observed in this study, and the results may not apply universally. Therefore, further research across diverse cultural contexts is needed to assess the broader applicability of these findings.

The synergies between luxury tourism and cultural tourism can enhance the overall travel experience by blending premium comfort with cultural authenticity, appealing to a diverse range of travelers looking for sophistication and meaningful connections in their journeys [82]. Therefore, an interesting direction for future research in this area would be to assess how individuals would respond to vacations that blended luxury tourism and cultural tourism. For example, it would be interesting to compare how narcissistic individuals responded to vacations that combined these forms of travel (e.g., staying at a luxurious hotel that was in close proximity to important cultural activities).

## 6. Conclusions

Despite the limitations of this research, it has expanded the current understanding of the appeal and preference for luxury tourism over cultural tourism activities by examining the role of personality traits. While luxury tourism is perceived favorably in terms of interest and intentions, cultural tourism may provide more profound, enriching experiences that lead to greater satisfaction. This distinction underscores the importance of understanding these preferences in tourism marketing. The contrast between the two also reveals the complex relationship between narcissistic personality traits and tourism preferences. Moreover, the current study proposed a possible mechanism by which travel-related motivations and preferences for activities mediate the connections that narcissistic personality traits have with the preference for luxury tourism over cultural tourism activities. More specifically, narcissistic personality traits were generally associated with the preference for luxury tourism over cultural tourism, with mainly indirect associations via the motives for relaxation and entertainment, status and social recognition, and experience. These results provide additional support for distinguishing between different aspects of narcissism when considering the connections that narcissism has with travel-related motivations and preferences for activities, and the preference for luxury tourism over cultural tourism.

In comparison to previous research, these findings align with studies indicating that narcissistic personality traits are often associated with a preference for luxurious experiences as a means of self-enhancement and status-seeking. Moreover, while these results show that luxury tourism is viewed positively in terms of interest and intention, they also support earlier findings indicating that cultural tourism provides more enriching and satisfying experiences. This study highlights the importance of distinguishing between different forms of narcissism and their distinct associations with travel preferences, further contributing to the understanding of how personality influences tourism behaviors.

## Figures and Tables

**Figure 1 behavsci-14-00972-f001:**
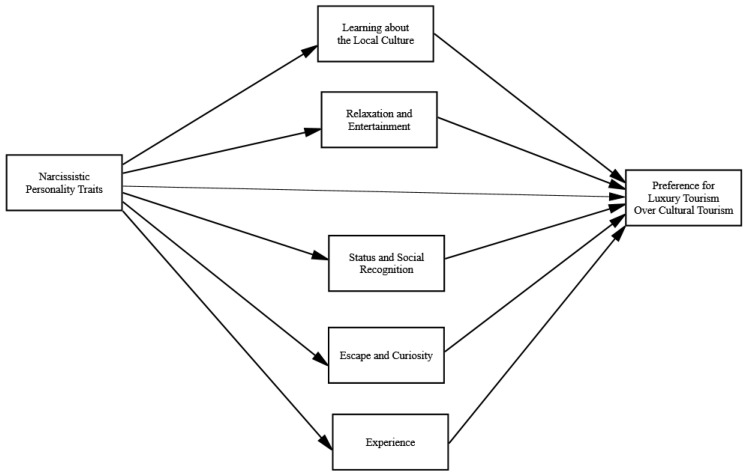
The proposed mediational model in which narcissistic personality traits have indirect associations with a preference for luxury tourism over cultural tourism activities through tourist motivations: a traveler’s need for learning about the local culture, relaxation and entertainment, status and social recognition, escape and curiosity, and experience.

**Figure 2 behavsci-14-00972-f002:**
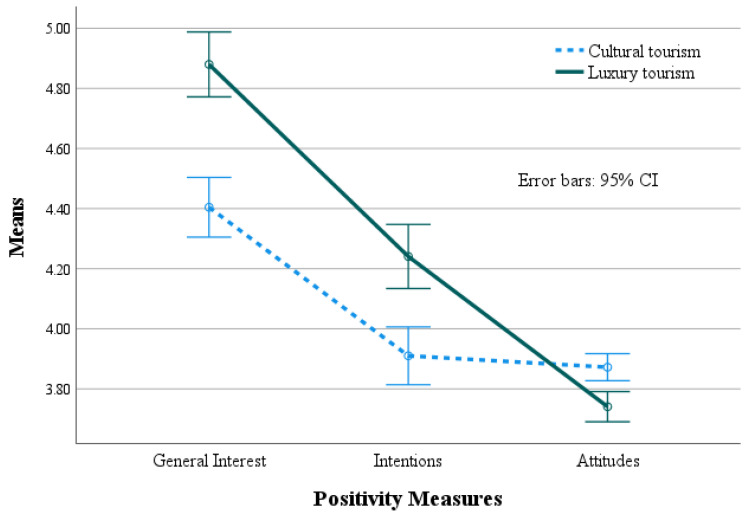
The interaction between the Type of Vignette and Positivity Measures.

**Table 1 behavsci-14-00972-t001:** M and SD for Measured levels of Positivity for Cultural and Luxury tourism.

	Type of Vignette							
	Cultural Tourism			Luxury Tourism			Post Hoc	
Positivity	M		SD	M		SD	Paired Samples t _[1341]_ Test	
1. General Interest	4.40 **^a^**		1.86	4.88 **^b^**		2.02	a ≠ b	−6.18 ***
2. Intentions	3.91 **^c^**		1.79	4.24 **^d^**		1.99	c ≠ d	−4.55 ***
3. Attitudes	3.87 **^e^**		0.84	3.74 **^f^**		0.93	e ≠ f	4.06 ***
**Post hoc** **Paired Samples t _[1341]_ test**	a ≠ c	18.91 ***		b ≠ d	22.14 ***			
	c ≠ e	0.96		d ≠ f	12.67 ***			

**** p* < 0.001.

**Table 2 behavsci-14-00972-t002:** Intercorrelations and Descriptive Statistics.

	1	2	3	4	5	6	7	8	9	10
1. Extraverted Narcissism	–									
2. Antagonistic Narcissism	0.43 ***	–								
3. Neurotic Narcissism	−0.02	0.10 ***	–							
4. Communal Narcissism	0.53 ***	0.32 ***	−0.04	–						
5. Motivation for Learning about the Local Culture	0.05	0.07 **	−0.06 *	0.17 ***	–					
6. Motivation for Relaxation and Entertainment	0.31 ***	0.11 ***	0.11 ***	0.28 ***	0.21 ***	–				
7. Motivation for Status and Social Recognition	0.27 ***	0.47 ***	0.15 ***	0.34 ***	0.24 ***	0.27 ***	–			
8. Motivation for Escape and Curiosity	0.19 ***	0.15 ***	0.14 ***	0.22 ***	0.28 ***	0.59 ***	0.18 ***	–		
9. Motivation for Experience	0.20 ***	0.28 ***	0.10 ***	0.27 ***	0.45 ***	0.52 ***	0.51 ***	0.34 ***	–	
10. Preference for Luxury Tourism Over Cultural Tourism	0.25 ***	0.21 ***	0.04	0.17 ***	−0.39 ***	0.25 ***	0.17 ***	0.07 *	0.12 ***	–
*Mean*	3.29	2.40	3.11	3.88	3.24	4.02	2.37	3.99	3.13	0.00
*Standard Deviation*	0.59	0.52	0.70	0.99	0.82	0.55	0.91	0.73	0.74	0.94

* *p* < 0.05; ** *p* < 0.01; *** *p* < 0.001.

**Table 3 behavsci-14-00972-t003:** Results of the Parallel Multiple Mediation Analyses.

	Extraverted Narcissism	Antagonistic Narcissism	Neurotic Narcissism	Communal Narcissism
** *Associations with Mediators* **				
Narcissism → Learning about the Local Culture	0.12 ***	0.13 ***	−0.03	0.24 ***
Narcissism → Relaxation and Entertainment	0.31 ***	0.11 ***	0.09 **	0.25 ***
Narcissism → Status and Social Recognition	0.27 ***	0.43 ***	0.17 ***	0.33 ***
Narcissism → Escape and Curiosity	0.17 ***	0.17 ***	0.11 ***	0.19 ***
Narcissism → Experience	0.22 ***	0.28 ***	0.10 ***	0.27 ***
** *Associations with Outcome* **				
Narcissism → Preference for Luxury Tourism Over Cultural Tourism (Total)	0.17 ***	0.13 ***	0.02	0.10 ***
Narcissism → Preference for Luxury Tourism Over Cultural Tourism (Direct)	0.11 ***	0.10 ***	−0.05 *	0.09 ***
Narcissism → Learning about the Local Culture → Preference for Luxury Tourism Over Cultural Tourism	−0.06 ***	−0.06 ***	0.01	−0.11 ***
Narcissism → Relaxation and Entertainment → Preference for Luxury Tourism Over Cultural Tourism	0.06 ***	0.02 ***	0.02 **	0.05 ***
Narcissism → Status and Social Recognition → Preference for Luxury Tourism Over Cultural Tourism	0.03 ***	0.03 ***	0.02 ***	0.03 ***
Narcissism → Escape and Curiosity → Preference for Luxury Tourism Over Cultural Tourism	0.00	0.00	0.00	0.00
Narcissism → Experience → Preference for Luxury Tourism Over Cultural Tourism	0.03 ***	0.04 ***	0.02 **	0.04 ***

* *p* < 0.05; ** *p* < 0.01; *** *p* < 0.001.

## Data Availability

The data presented in this study are available on request from the corresponding authors.

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
