# Peer review of "Luxury or Cultural Tourism Activities? The Role of Narcissistic Personality Traits and Travel-Related Motivations"

_behavsci, 2024, doi:10.3390/bs14100972_

Round 1

Reviewer 1 Report (Previous Reviewer 2)

Comments and Suggestions for Authors

While I appreciate the effort the authors have put into addressing the feedback provided in the initial review, I must express my concerns regarding the extent to which these revisions have resolved the fundamental issues previously raised. In other words, I worry that the current version still falls short of adequately addressing the fundamental concerns about the manuscript's contribution to the field. 

Another thing I want to ask is why there were so many edits on the stats in the results section. Does that mean the author's original analysis was incorrect? I am a bit confused.

Comments on the Quality of English Language

None.

Author Response

Reviewer 1:

"While I appreciate the effort the authors have put into addressing the feedback provided in the initial review, I must express my concerns regarding the extent to which these revisions have resolved the fundamental issues previously raised. In other words, I worry that the current version still falls short of adequately addressing the fundamental concerns about the manuscript's contribution to the field." 

Response: Thank you for your feedback and for acknowledging our efforts in addressing the initial concerns raised in your previous review. We appreciate your continued engagement with our work, even though we understand that our revisions may not have fully addressed your fundamental concerns about the manuscript’s contribution. With respect to the manuscript's contribution to the field, we would like to refer the Academic Editor to the other reviewer that in the first round (Reviewer 3 who specified that they were a tourism scholar) indicated: “The paper addresses a very interesting and novel topic. As a tourism scholar, it is always good to see transdisciplinary work on tourism issues coming from scholars of different areas, and psychology plays a great role in understanding tourists’ (as consumers’ in general) choices and behaviour." And now in the second round indicated again (in this round they are Reviewer 2): "Once again, this is a very interesting transdisciplinary research piece with some insightful findings."…..

"Another thing I want to ask is why there were so many edits on the stats in the results section. Does that mean the author's original analysis was incorrect? I am a bit confused."

Response: Thank you for your inquiry regarding the edits in the results section. These changes were made in accordance with the Academic Editor's request to re-analyze the models while controlling for demographic variables. This process was intended to enhance the accuracy of our analyses and provide a clearer understanding of the findings. We appreciate your attention to this matter, and we remain open to any specific suggestions you might have for further improving the manuscript.

Reviewer 2 Report (Previous Reviewer 3)

Comments and Suggestions for Authors

Once again, this is a very interesting transdisciplinary research piece with some insightful findings. My only problem is still with the organization/presentation. This has indeed been much improved from the original manuscript. Some parts of the literature are more developed and clarified, particularly the specific aspects of communal narcissism and its potential relationship with tourism consumption preferences. Now you also describe the hypotheses, justifying them with points from the literature you have addressed. But here also lies the main flaw of the current manuscript:

1.      You raise the hypotheses, present the results, but do not mention within the results, whether and how they support the hypotheses. You don’t even mention the hypotheses anywhere in the work after presenting them. This needs to be addressed and can be easily done. Simply state, after the piece of result that supports (or does not) each hypothesis: “this supports Hx”.

2.      Also, Figure 1, which was ok in the original manuscript, now is not readable. Please, provide a version of the figure with better resolution/readability.

These are the only aspects that must necessarily be addressed, and the only reasons I am once again requiring minor reviews. However, there are other suggestions that, should the authors decide to accept, would greatly improve the papers’ presentation:

1.      The literature review should be outside the Introduction (i.e., be the chapter 2, not 1.3). Having such a big introduction, almost half the paper, might not be outright wrong, but just seems unnecessarily unorthodox.

2.      The tittle of the section “hypotheses development” is inaccurate, as you do not really develop the hypotheses there, but simply address the previous contributions that are relevant to them. Therefore, it’s simply literature review. Either leave the literature review in a single section, or divide it in the sections that are currently 1.2 and 1.2.1 with tittles that reflect their contents (e.g., TPB and tourism motives and; Personality traits and preferences for cultural and luxury tourism).

I really hope the authors address at least the most critical issues and that the paper is published as soon as possible. 

Author Response

Reviewer 2:

"Once again, this is a very interesting transdisciplinary research piece with some insightful findings. My only problem is still with the organization/presentation. This has indeed been much improved from the original manuscript. Some parts of the literature are more developed and clarified, particularly the specific aspects of communal narcissism and its potential relationship with tourism consumption preferences. Now you also describe the hypotheses, justifying them with points from the literature you have addressed."

Response: Thank you for your continued support and positive feedback throughout the review process. Thank you very much for your kind support and insightful comments. We appreciate your recognition of the improvements made in the organization and presentation of the manuscript, as well as your acknowledgment of the developments regarding communal narcissism and its relationship with tourism consumption preferences. Your constructive feedback has been invaluable in refining our hypotheses and the corresponding literature review. I look forward to further enhancing the manuscript based on your suggestions.

But here also lies the main flaw of the current manuscript:

  1. "You raise the hypotheses, present the results, but do not mention within the results, whether and how they support the hypotheses. You don’t even mention the hypotheses anywhere in the work after presenting them. This needs to be addressed and can be easily done. Simply state, after the piece of result that supports (or does not) each hypothesis: “this supports Hx”."

Response: Thank you for your valuable suggestion. We have now incorporated explicit statements in the results section indicating whether each hypothesis is supported, partially supported, or if a more complex insight emerged. This addition enhances clarity and directly addresses your concern.

  1. "Also, Figure 1, which was ok in the original manuscript, now is not readable. Please, provide a version of the figure with better resolution/readability."

Response: Thank you for bringing this to our attention. Although we did not replace Figure 1 and it remains the same as in the original manuscript, we have enhanced its resolution and readability. We will ensure that a higher-quality version of the figure is provided to enhance clarity. We suspect the problem is caused by the fact that the PDF contained edits, and the comments may have altered the size of the page.

  1. "These are the only aspects that must necessarily be addressed, and the only reasons I am once again requiring minor reviews. However, there are other suggestions that, should the authors decide to accept, would greatly improve the papers’ presentation:

  • The literature review should be outside the Introduction (i.e., be the chapter 2, not 1.3). Having such a big introduction, almost half the paper, might not be outright wrong, but just seems unnecessarily unorthodox."

Response: Thank you for your feedback. We have restructured the manuscript so that the literature review is now presented as Section 2, with all subsequent numbering adjusted accordingly. We appreciate your suggestion, as it significantly enhances the paper’s presentation.

  • "The tittle of the section “hypotheses development” is inaccurate, as you do not really develop the hypotheses there, but simply address the previous contributions that are relevant to them. Therefore, it’s simply literature review. Either leave the literature review in a single section, or divide it in the sections that are currently 1.2 and 1.2.1 with tittles that reflect their contents (e.g., TPB and tourism motives and; Personality traits and preferences for cultural and luxury tourism)."

Response: Thank you for your insightful suggestion. In response, we have removed the title for the “Hypotheses Development” section and integrated it into the literature review, resulting in a single cohesive section. This change enhances the clarity and flow of the manuscript.

Reviewer 3 Report (New Reviewer)

Comments and Suggestions for Authors

This study aimed to investigate the associations that narcissistic personality traits have concerning the preference for luxury tourism over cultural tourism, considering travel-related motivations as potential mediating factors.

 Title: The title is clear and reflects the article's content according to a case study.

Abstract: The abstract does not clearly and concisely present the main points of the article - objective, methodology, results and conclusions - so it should be improved.

Keywords: Keywords can be expanded to better index the article.

Introduction: The introduction contextualizes the theme and defines the problem, with the objectives of the work highlighted in another subtitle.

Methodology - The methodology used is appropriate and is described clearly and in detail, allowing replication. The methods used guarantee the validity and reliability of the results.

Presentation of the results and Discussion: The discussion of the results is adequate, but the results are presented in a somewhat confusing way and with too many changes that complicate the perception of them.  It would be interesting to compare this with other previous studies.

Conclusion: The conclusion summarizes the article's main findings, indicating that luxury tourism was perceived more favourably than cultural tourism in terms of general interest and intentions. Still, cultural tourism was perceived more favourably than luxury tourism in terms of attitudes. Although they have pointed out the difficulties and limitations, this item can be improved or not exist and be included in the discussion.

The article's structure is organized logically, but so many subtitles are unnecessary, and the literature review should be an independent point.

The development of the text does not always flow well between paragraphs. The fact that they submitted a manuscript with several corrections that we do not always understand does not help the understanding of it either.

Figure 1 should be improved because it cannot be understood.

Comments on the Quality of English Language

Moderate editing of English language required.

Author Response

Reviewer 3:

"This study aimed to investigate the associations that narcissistic personality traits have concerning the preference for luxury tourism over cultural tourism, considering travel-related motivations as potential mediating factors.

 Title: The title is clear and reflects the article's content according to a case study."

Response: Thank you for acknowledging that the title clearly reflects the content of our article. We aimed to ensure that the title accurately conveys the focus of our study, which investigates the associations between narcissistic personality traits and the preference for luxury tourism over cultural tourism, with travel-related motivations as potential mediating factors. We appreciate your positive feedback on this aspect, and we are committed to maintaining clarity and alignment throughout the manuscript.

"Abstract: The abstract does not clearly and concisely present the main points of the article - objective, methodology, results and conclusions - so it should be improved."

Response:  Thank you for your valuable feedback regarding the abstract. We understand the importance of clearly presenting the main points of the article, including the objective, methodology, results, and conclusions. In response to your comment, we have revised the abstract to enhance its clarity and conciseness. The revised abstract now includes the following improvements: Objective: We explicitly state the aim of the study, emphasizing the role of travel-related motivations and preferences for activities in the context of narcissistic personality traits and tourism preferences. Methodology: We clearly outline the sample size and analytical approach (parallel mediation analyses) used in the study, providing a more precise overview of our methods. Results: The updated abstract summarizes the significant findings related to the associations between narcissistic traits and preferences for luxury versus cultural tourism, highlighting the mediation of various travel motivations. Conclusions: We conclude with a statement that reflects the broader implications of our findings for tourism management strategies, particularly in understanding how personality influences tourism preferences. We believe that these adjustments address your concerns and significantly enhance the abstract's effectiveness. Thank you again for your insights, which have helped improve the clarity of our manuscript.

"Keywords: Keywords can be expanded to better index the article."

Response: Thank you for your suggestion to expand the keywords for better indexing. In response, we have revised our keyword list to more effectively capture the primary themes and concepts of the study. We have added the terms "Personality" and "Travel," as well as the keywords "Activity Preferences" and "Tourism Management." These additions aim to enhance the article's accessibility and discoverability for interested readers. We appreciate your valuable feedback and are committed to making necessary improvements.

"Introduction: The introduction contextualizes the theme and defines the problem, with the objectives of the work highlighted in another subtitle."

Response: Thank you for your feedback on the introduction. We appreciate your recognition that the introduction effectively contextualizes the theme and defines the problem. In response to suggestions from previous reviews, we have highlighted the objectives of the work using a separate subtitle to enhance clarity and focus. We are open to further suggestions on how we might improve this section and sincerely appreciate your input.

"Methodology - The methodology used is appropriate and is described clearly and in detail, allowing replication. The methods used guarantee the validity and reliability of the results."

Response: Thank you for your positive feedback regarding our methodology. We are glad to hear that you find it appropriate, clearly described, and sufficiently detailed to allow for replication. Ensuring the validity and reliability of our results is of utmost importance to us, and we appreciate your acknowledgment of this aspect of our work.

"Presentation of the results and Discussion: The discussion of the results is adequate, but the results are presented in a somewhat confusing way and with too many changes that complicate the perception of them.  It would be interesting to compare this with other previous studies."

Response: Thank you for your feedback on the presentation and discussion of our results. We appreciate your positive assessment of the discussion's adequacy. We understand that the clarity of the results' presentation may have been affected by corrections, which were made in response to the Academic Editor’s requests during a previous round of revisions. We have worked on refining this section to present the findings more clearly and coherently.

Additionally, we have enhanced our comparison with previous studies to provide broader context and deepen the understanding of our findings. Your insights have been invaluable in guiding these improvements.

"Conclusion: The conclusion summarizes the article's main findings, indicating that luxury tourism was perceived more favourably than cultural tourism in terms of general interest and intentions. Still, cultural tourism was perceived more favourably than luxury tourism in terms of attitudes. Although they have pointed out the difficulties and limitations, this item can be improved or not exist and be included in the discussion."

Response: Thank you for your valuable feedback regarding the conclusion section of our paper. We have carefully considered your suggestions and made the following revisions to enhance clarity and address the points you raised. We have clarified the summary of the main findings, emphasizing the dual perceptions of luxury and cultural tourism. Specifically, we now clearly state that luxury tourism is perceived more favorably in terms of interest and intentions, while cultural tourism offers deeper, enriching experiences and satisfaction. We have incorporated a more explicit discussion of the study's limitations, emphasizing how these limitations inform our understanding of the findings and their implications. We revised the conclusion to highlight the proposed mechanisms through which travel-related motivations interact with narcissistic personality traits and preferences for tourism types. We ensured that our discussion reinforces the need to differentiate between various forms of narcissism and their implications for travel behavior, contributing to the wider discourse on personality in tourism. We believe these changes address your concerns and improve the overall quality of the conclusion. Thank you once again for your constructive comments, which have been incredibly helpful in refining our manuscript.

"The article's structure is organized logically, but so many subtitles are unnecessary, and the literature review should be an independent point."

Response:

Thank you for your feedback regarding the article's structure. The additional subtitles in the introduction (literature review) were included in response to a specific request from Reviewer 1 in the previous round of reviews. We aimed to enhance clarity and organization through this adjustment. Regarding the literature review, we restructured it as an independent section with no subtitles to improve coherence and readability. We appreciate your insights and suggestions as they help us refine our work.

"The development of the text does not always flow well between paragraphs."

Response: Thank you for your observation about the flow of the text between paragraphs. We updated some of the transitions to ensure a more coherent and seamless progression throughout the manuscript. Your feedback has been extremely helpful for us in terms of improving the clarity and readability of our work.

"The fact that they submitted a manuscript with several corrections that we do not always understand does not help the understanding of it either."

Response: Thank you for your feedback. We recognize that the manuscript underwent numerous revisions, which may have impacted its clarity. These edits were made in response to requests from Reviewer 1 in the previous round of review, for changes to the structure of the introduction and the addition of explicit hypotheses. Additionally, changes were made at the request of the Academic Editor to re-analyze the models while controlling for demographic variables. These revisions were intended to improve the accuracy of our analyses and clarify our findings. We appreciate your attention to these details and are open to any specific suggestions you might have for further improvement.

"Figure 1 should be improved because it cannot be understood."

Response:

Thank you for your valuable feedback regarding Figure 1. We understand the importance of clarity in visual representations and will take steps to enhance the figure to ensure it is easily understandable.  To address this, we have submitted the figure as a separate JPG file in addition to its inclusion in the main document. We hope this will enhance clarity and readability. Please let us know if there are any further adjustments needed. However, we suspect the problem was caused by the fact that the PDF contained edits, and the comments impacted the page size

Reviewer 4 Report (New Reviewer)

Comments and Suggestions for Authors

It was a pleasure to read the paper. Interesting topic.

I draw attention to the fact that you do not use the word we/our (line 198) but authors

Figure 1 is not clear, not readable

In what period was the research conducted? Which statistical package is corrupted?

I don't see which earlier works were the model for the research.

In conclusion, compare the obtained results with earlier research.

Author Response

Reviewer 4:

" It was a pleasure to read the paper. Interesting topic."

Response: Thank you for your kind words and for finding our paper enjoyable to read. We are pleased to hear that you found the topic interesting. Your positive feedback is greatly appreciated and encourages us in our research efforts.

"I draw attention to the fact that you do not use the word we/our (line 198) but authors"

Response:  Thank you for highlighting the inconsistency in the language use within our manuscript. We have revised the document, replacing all instances of "we/our" with passive voice constructions to maintain consistency. We appreciate your attention to detail and your helpful guidance in improving our paper.

"Figure 1 is not clear, not readable"

Response: Thank you for your feedback regarding Figure 1. We apologize for any issues with its readability. To address this, we have submitted the figure as a separate JPG file in addition to its inclusion in the main document. We hope this will enhance clarity and readability. Please let us know if there are any further adjustments needed. We suspect the problem was caused by the fact that the PDF contained edits, and the comments impacted the page size

"In what period was the research conducted?"

Response:  Thank you for your inquiry about the research timeline. The study was conducted over approximately 2.5 months (76 days), specifically from April 15 to the end of June 2024. We began data collection on the day we received IRB ethical approval [As noted in the manuscript: protocol code #00490 [15.04.2024]). We hope this information is helpful and are happy to provide further details if needed.

"Which statistical package is corrupted?"

Response: Thank you for your inquiry. To clarify, we have ensured that the Statistical Analyses section indicates that the MANOVA and correlations were conducted using SPSS version 27 (SPSS Inc., Chicago, IL). As mentioned, the mediational models were analyzed using the PROCESS macro for a series of parallel multiple mediation analyses. We have checked and confirmed that no statistical package used in our analyses is corrupted; you may have intended to ask about the software we incorporated. We appreciate your attention to this detail and are ready to address any further concerns.

"I don't see which earlier works were the model for the research."

Response:  Thank you for your valuable comment regarding the earlier works that served as models for our research. We appreciate your request for clarification. In our manuscript there is a section in the Literature Review that explicitly cites foundational studies that inform our approach. Notably, we reference works such as Besser and Priel (2006, [1]), which explore the influence of personality traits on leisure activities, and recent studies on luxury consumption and tourism behavior, including Kim (2018) ([12]) and Jain et al. (2023, [8]). These and other references provide a theoretical basis for examining how narcissistic personality traits influence preferences for luxury tourism over cultural tourism. We hope that it clarifies the conceptual framework guiding our research and adequately addresses your concerns.

"In conclusion, compare the obtained results with earlier research."

Response:  Thank you for your insightful suggestion to compare our obtained results with earlier research in the conclusion. We have revised the Discussion section to include a discussion that highlights how our findings align with and contribute to existing literature. Specifically, we note that our results resonate with previous studies indicating that narcissistic traits often lead to preferences for luxurious experiences as a means of self-enhancement and status-seeking. Furthermore, we contrast our findings regarding the positive perceptions of luxury tourism against the more enriching experiences typically associated with cultural tourism. By incorporating these comparisons, we aim to provide a more comprehensive understanding of how our study fits into the broader context of tourism research. We appreciate your feedback which has helped enhance the depth of our analysis.

Round 2

Reviewer 1 Report (Previous Reviewer 2)

Comments and Suggestions for Authors

I appreciate the authors' efforts in revising the manuscript and addressing my previous comments. I believe the revised manuscript is much improved and close to being ready for publication.

This manuscript is a resubmission of an earlier submission. The following is a list of the peer review reports and author responses from that submission.

Round 1

Reviewer 1 Report

Comments and Suggestions for Authors

Dear authors,

I have the pleasure to review your article.

“Luxury tourism emphasizes opulence and exclusivity” (row 78), field that is constantly developing, requires creativity and creates quality.

The market segment for luxury tourism is different, periodically goes through transformations and this aspect is well defined in this work.

The motivations for this type of tourism are multiple, it depends a lot on the individual personality traits, they are very clearly presented in the paper.

I find interesting the attention to study the touristic motivation of narcissistic people. Also, the motivations compared between luxury tourism and cultural tourism is opportune.

The research methodology is adequate and the discussions are detailed and clear.

The results of the research show the preferences of each type of narcissist, indicating a direction for the suppliers who prepare the offers in the field of luxury and cultural tourism.

The paper is well structured and easy to understand. The literature is rich with 79 references, edited according to requirements.

Sincerly, 

Author Response

Reviewer 1 Comments: “I have the pleasure to review your article. “Luxury tourism emphasizes opulence and exclusivity” (row 78), field that is constantly developing, requires creativity and creates quality. The market segment for luxury tourism is different, periodically goes through transformations and this aspect is well defined in this work. The motivations for this type of tourism are multiple, it depends a lot on the individual personality traits, they are very clearly presented in the paper. I find interesting the attention to study the touristic motivation of narcissistic people. Also, the motivations compared between luxury tourism and cultural tourism is opportune. The research methodology is adequate, and the discussions are detailed and clear. The results of the research show the preferences of each type of narcissist, indicating a direction for the suppliers who prepare the offers in the field of luxury and cultural tourism. The paper is well structured and easy to understand. The literature is rich with 79 references, edited according to requirements.”

Response: We would like to express our sincere gratitude for your thoughtful and constructive evaluation of our paper. Your positive feedback regarding the clarity of our presentation, the relevance of our research methodology, and the detailed discussions we provided is incredibly encouraging.

We appreciate your recognition of the distinct market segment of luxury tourism and the importance of creativity within this constantly evolving field. It is gratifying to know that our exploration of the motivations behind luxury tourism, particularly among individuals with narcissistic traits, resonated with you.

Your acknowledgment of the comparisons made between luxury tourism and cultural tourism highlights a critical facet of our research, and we are pleased to hear that you found it relevant and timely. We are also glad that you found our structured approach and rich literature review satisfactory.

Thank you once again for your insightful comments and for taking the time to review our work.

Reviewer 2 Report

Comments and Suggestions for Authors

The topic about luxury tourism and cultural tourism is interesting, but the specific research question that this paper looked at and the findings don't seem to be very informative or exciting. First, it's unclear why the authors depict luxury tourism against cultural tourism since these two types of tourism are not mutually exclusive. Second, while it's nice to look at the different types of narcissism, I didn't see a compelling story of how they will influence the choice of luxury vs. cultural tourism differently. The results also showed that three of the four types had positive correlation with luxury over cultural tourism and one nonsignificant correlation. So the general take away is that there might not be too much difference among the different types of narcissism on preference for luxury over cultural tourism. Third, the mixed findings on general interest, intentions, and attitude are a bit weirld. Theoretically, they should work the same. Finally, with one survey and correlational analysis, it's hard to make any convincing conclusions, especially with the process(mediation). I would suggest the authors to conduct some experiments to provide further evidence.

Comments on the Quality of English Language

English language is fine but the writing could be improved.

Author Response

Reviewer 2 Comments: “The topic about luxury tourism and cultural tourism is interesting, but the specific research question that this paper looked at and the findings don't seem to be very informative or exciting. First, it's unclear why the authors depict luxury tourism against cultural tourism since these two types of tourism are not mutually exclusive. Second, while it's nice to look at the different types of narcissism, I didn't see a compelling story of how they will influence the choice of luxury vs. cultural tourism differently. The results also showed that three of the four types had positive correlation with luxury over cultural tourism and one nonsignificant correlation. So the general take away is that there might not be too much difference among the different types of narcissism on preference for luxury over cultural tourism. Third, the mixed findings on general interest, intentions, and attitude are a bit weird. Theoretically, they should work the same. Finally, with one survey and correlational analysis, it's hard to make any convincing conclusions, especially with the process(mediation). I would suggest the authors to conduct some experiments to provide further evidence.”

Response:  Thank you for your thorough and thoughtful review of our manuscript. We appreciate your insights and understand your concerns regarding the clarity and implications of our research findings.

  1. Depiction of Luxury Tourism vs. Cultural Tourism:
    You rightly point out that luxury and cultural tourism are not mutually exclusive. Our aim in contrasting these two types of tourism was to highlight the different motivational aspects and experiences they offer, specifically in the context of narcissism. We have revisited the introduction according to |Reviewer’s 3 suggestions. We feel that the introduction clarifies our rationale for this comparison, emphasizing the unique attributes and experiential outcomes of each type, which we believe enrich our understanding of consumer motivations. Nevertheless, as already stated in the discussion section: “…an interesting direction for future research in this area would be to assess how individuals would respond to vacations that blended luxury tourism and cultural tourism. For example, it would be interesting to compare how narcissistic individuals responded to vacations that combined these forms of travel (e.g., staying at a luxurious hotel that was in close proximity to important cultural activities).”
  2. Influence of Narcissism on Tourism Preferences:
    We acknowledge your point regarding the nuances of how different types of narcissism may or may not influence the preference for luxury versus cultural tourism. We believe the revised introduction and the hypotheses development address this by elaborating on our discussion of the different types of narcissism. As can be seen it is emphasized that while our results indicate a general preference for luxury tourism, this does not negate the potential diverse motivations that may exist (we also add specific predictions/hypotheses as recommended by Reviewer 3. We agree that the relationship is complex, and thus the discussion contained implications of our findings in the context of existing literature.
  3. Mixed Findings on Interests, Intentions, and Attitudes:
    Your comment regarding the mixed findings is well-taken. Our robust theoretical framework addresses these inconsistencies, explores potential reasons for these results and discusses their implications for future research.
  4. Methodological Limitations:
    We appreciate your critique of our methodological approach. While we recognize that correlational analysis presents certain limitations, we believe that it provides valuable insight into preliminary relationships and trends. In the revised manuscript, we acknowledge these limitations more explicitly and recommend potential future research to use experimental and longitudinal designs to build on our findings. We appreciate your suggestion for further research and will explore these avenues in future work

We hope that the revisions we have made address your concerns and enhance the overall quality of our paper. Thank you once again for your constructive feedback; it has been invaluable in refining our work.

Reviewer 3 Report

Comments and Suggestions for Authors

The paper addresses a very interesting and novel topic. As a tourism scholar, it is always good to see transdisciplinary work on tourism issues coming from scholars of different areas, and psychology plays a great role in understanding tourists’ (as consumers’ in general) choices and behaviour.

The literature review is robust, and the methods are well defined and executed. They are also well reported, with one minor flaw regarding the explanation of the sampling method (addressed in the attached file). The only problems I see with the manuscript are minor, and can be easily addressed. The first and main problem has to do with the clarity of the Introduction. The introduction would benefit from a more structured approach. I understand you aimed for the more direct approach of including the essential theoretical overview in the introduction. However, you have too much literature to cover, and end up going back and forth with the contents of the introduction. For instance, you mention aspects of the methods before stating the objective. Then, after stating it, you go back to literature review. Considering this, I believe separating between Introduction and Literature Review (or however you want to call it) would help the author(s) better structure their exposition.

In the same vein, the work would benefit from a more structured exposition of the conceptual model. Eg. "Literature has shown that (...) therefore, H1: ...". Considering my previous comment, this could come throughout the literature review. I understand it quite complicated in the case of this study, as you test one model for each narcissism trait, but you should try and make it more structured.

Finally, there are several typos, mainly missing punctuation, especially in the Discussion section. These are all highlighted in the attached file.

Author Response

Reviewer 3 Comments: “The paper addresses a very interesting and novel topic. As a tourism scholar, it is always good to see transdisciplinary work on tourism issues coming from scholars of different areas, and psychology plays a great role in understanding tourists’ (as consumers’ in general) choices and behaviour.

The literature review is robust, and the methods are well defined and executed. They are also well reported, with one minor flaw regarding the explanation of the sampling method (addressed in the attached file). The only problems I see with the manuscript are minor and can be easily addressed.

The first and main problem has to do with the clarity of the Introduction. The introduction would benefit from a more structured approach. I understand you aimed for the more direct approach of including the essential theoretical overview in the introduction. However, you have too much literature to cover, and end up going back and forth with the contents of the introduction. For instance, you mention aspects of the methods before stating the objective. Then, after stating it, you go back to literature review. Considering this, I believe separating between Introduction and Literature Review (or however you want to call it) would help the author(s) better structure their exposition.

In the same vein, the work would benefit from a more structured exposition of the conceptual model. Eg. "Literature has shown that (...) therefore, H1: ...". Considering my previous comment, this could come throughout the literature review. I understand it quite complicated in the case of this study, as you test one model for each narcissism trait, but you should try and make it more structured.

Finally, there are several typos, mainly missing punctuation, especially in the Discussion section. These are all highlighted in the attached file.”

Response:  Thank you for your thoughtful and encouraging review of our manuscript. We appreciate your recognition of the novelty of the topic and the value of a transdisciplinary approach in understanding tourist behavior. Your feedback is invaluable, and we are grateful for your insightful comments.

We acknowledge your concern regarding the structure of the Introduction. We agree that a clearer separation between the Introduction and the Literature Review would enhance the overall flow and clarity of the paper. In the revised manuscript, we have restructured these sections to clearly define the objective of our study before delving into the relevant literature. This separation will help readers grasp the foundational concepts before engaging with the specific contributions of our research. As can be seen we have divided the previous section named “introduction” to subsections:

  1. Introduction; 1.1 Objectives; 1.2. Literature Review; 1.2.1 Hypotheses Development; and 1.3 Overview and Predictions – in this sub-section we have presented the proposed hypotheses.

We highly appreciate your meticulous attention to detail regarding the typos and missing punctuation, particularly in the Discussion section as well as the comments you have generously added to the manuscript. We have carefully reviewed the manuscript and corrected all highlighted issues to ensure improved readability and professionalism.

We believe that these revisions will substantially improve the clarity and quality of the manuscript.

Thank you once again for your constructive feedback, which will help us refine our work further.
